# Concentrations of Progesterone (P_4_), Anti-Müllerian Hormone (AMH), and Haptoglobin (Hp) in Pregnant and Non-Pregnant Ewes and Their Association with Fetal Mortality, Maternal Weight, and Twinning Rate

**DOI:** 10.3390/vetsci12050463

**Published:** 2025-05-12

**Authors:** Halil Gunes Ozturan, Selim Aslan, Feride Zabitler Tepik, Isfendiyar Darbaz, Serkan Sayiner, Axel Wehrend

**Affiliations:** 1Department of Obstetrics and Gynecology, Faculty of Veterinary Medicine, Near East University, Nicosia 99138, Cyprus; selim.aslan@neu.edu.tr (S.A.); feride.zabitler@neu.edu.tr (F.Z.T.); 2DESAM Research Institute, Near East University, Nicosia 99138, Cyprus; 3Department of Biochemistry, Faculty of Veterinary Medicine, Near East University, Nicosia 99138, Cyprus; serkan.sayiner@neu.edu.tr; 4Veterinary Clinic for Reproductive Medicine and Neonatology, Justus-Liebig-University, 35392 Giessen, Germany; wehrend@vetmed.uni-giessen.de

**Keywords:** sheep, ewe, pregnancy, single-twin gestation, body weight, progesterone, anti-müllerian hormone, haptoglobin

## Abstract

In sheep, the application of the Progesterone (P_4_) test enables not only the distinction between pregnant (P+) and non-pregnant (P−) animals but also the detection of fetal deaths (FD). P_4_ levels are a significant parameter for identifying fetal death as well as distinguishing between single and twin pregnancies. Higher Anti-Müllerian Hormone (AMH) levels observed up to day 30 of pregnancy compared to FD cases may serve as an early indicator of fetal viability. Marked decreases in AMH and haptoglobin (Hp) levels toward the end of pregnancy are important for monitoring the later stages of gestation. Furthermore, both AMH and Hp levels vary according to seasonal periods and maternal body weight (BW) changes. P_4_ levels were also found to be associated with BW. Collectively, these findings suggest that biochemical markers such as P_4_, AMH, and Hp play a critical role in monitoring pregnancy progression, fetal development, and physiological responses to seasonal and body weight changes in sheep.

## 1. Introduction

Sheep breeding holds an important place in animal farming and provides significant contributions to national economies and public health. The increase in sheep populations is largely driven by higher global meat and dairy consumption, making sheep one of the most important livestock for the economy and industry due to their high production and reproduction characteristics [1]. However, early recognition of pregnancy and the determination of changes in the pregnancy process are essential due to the low number of modern breeding facilities and the fact that most breeding is pasture-based [2].

The determination of litter size in ewes is vital for pregnancy management. Particularly, during the final 4–6 weeks of gestation, appropriate dietary rationing is essential for the health of the mother for adequate development of the fetuses in the uterus and for a healthy birth weight. Underfeeding or reduced food intake can lead to the birth of weak offspring and metabolic diseases such as pregnancy toxemia. Therefore, adequate and balanced diets can prevent pregnancy toxemia, ensure appropriate birth weight, and increase the survival rate of offspring in multiple pregnancies [3,4].

Despite extensive research on domestic animals’ reproductive health, embryonic mortality remains a significant issue, causing losses in both commercial breeding and scientific research. Although there is limited information on early fetal mortality in ewes, the rate of early fetal mortality varies between 3.5% and 12%. The diagnosis of pregnancy in ewes is possible with 100% accuracy through a frequency of 3.5 MHz by trans abdominal route from day 40 [2,5].

Haptoglobin (Hp) belongs to the group of plasma proteins of hepatic origin called acute-phase proteins (APPs). The acute phase response is a physiological reaction that occurs in the first days following tissue injury and/or infection. In veterinary medicine, the determination of APP concentrations can provide useful information confirming the presence of an active inflammatory process. Studies conducted in sheep and cattle have concluded that a Hp concentration exceeding 0.4 g/L is indicative of infection, while a level around 0.2 g/L may suggest an early or mild infection [6,7,8]. The concentration of APPs has been investigated in various species such as sheep, goats, dogs, and pigs; it is typically low in the plasma of healthy animals but can increase by 100 to 1000 times in response to inflammation. APP levels then decrease rapidly within 24 to 48 h following injury and during recovery. Some APPs are directly involved in the immune reaction of the animal, while others have the effect of protecting the tissue from pathogens and enzymes secreted by phagocytic cells [9,10]. In dairy cattle, alterations in Hp concentrations have been consistently linked to various periparturient disorders, including dystocia, puerperal metritis, retained placenta, and metabolic stress characterized by elevated β-hydroxybutyrate levels [11,12]. Importantly, Huzzey et al. proposed that Hp profiling may serve as a valuable biomarker for the early identification of metritis, potentially improving the efficacy of therapeutic and preventive strategies [11].

Anti-Müllerian Hormone (AMH) is a dimeric glycoprotein and a member of the transforming growth factor-β (TGF-β) superfamily. It is produced by the granulosa cells of early antral follicles in the ovaries. Pursuant to some studies, circulating blood AMH concentrations are a reliable endocrine marker for the size of the antral follicle population and a key indicator of fertility [13]. In pregnant cows, a significant correlation was found between the number of small antral follicles counted by ovarian USG and plasma AMH concentrations. Thus, these results suggest that the changes in AMH concentrations observed during the stage from pregnancy to the postpartum period are the result of numerical changes in the population of follicles secreting high AMH in the ovaries [14,15].

The concentration of progesterone (P_4_) in peripheral plasma rises gradually during the luteal phase in the first half of pregnancy, increases markedly at about 90 days post-mating, with concentrations peaking at 125 days post-mating and declining in the last few days before parturition. In sheep, during the first third of gestation, P_4_ is produced by the corpus luteum (CL); production is taken over by the placenta after about 50 days post-insemination, and removal of the ovaries after this day does not jeopardize fetal development [16,17].

Obesity has been linked to increased blood Hp concentrations, as demonstrated in several studies. During pregnancy, serum Hp concentrations follow a biphasic pattern, with peaks occurring in the first and third trimesters. Notably, infertile women have been found to have lower levels of naturally occurring antibodies against Hp compared to fertile controls, indicating a possible association between immune response and reproductive function [18].

In recent years, AMH has emerged as an important biomarker in reproductive medicine. Evidence indicates that AMH levels may predict twin pregnancies following assisted reproduction, supporting its role in assessing oocyte and embryo quality [19]. Several studies have further highlighted the association between AMH concentrations and embryo quality [20,21,22], as well as higher clinical pregnancy [23,24,25] and live birth rates [26].

Moreover, a significant relationship has been established between obesity, body mass index, and AMH concentrations in women [27], indicating that metabolic status may impact ovarian reserve. Additionally, the number of fetuses has been found to correlate with progesterone levels, a hormone known to protect against embryonic loss and support embryonic development [28,29,30].

This study aims to determine the changes in Hp, AMH, and P_4_ concentrations and the differences in P+, P−, and FD cases determined by USG examinations during pregnancy in sheep. Additionally, the study aims to reveal the relationships between these parameters and phenomena such as maternal weight and twins.

## 2. Materials and Methods

### 2.1. Animals, Examinations, and Sampling

In this study, a total of 39 Assaf ewes, each 3 years old and with no reproductive problems, were used. The animals were raised on a 3000-head farm, located in the Nicosia region of Northern Cyprus. As part of the experimental protocol, an estrus synchronization program was implemented during the breeding season. This program involved the use of Chronogest^®^ CR (20 mg controlled-release intravaginal sponge), specifically designed for sheep. Following synchronization, all ewes were subjected to controlled and simultaneous natural mating with rams in order to ensure uniformity in the reproductive cycle. Day zero was determined as the day of mating, and the weight of the ewes was measured on days 10, 80, and 150. Blood samples were collected (from vena jugularis) on days 10, 20, 30, 40, 60, 80, and 150 after mating, and serum samples were stored at −20 °C until analysis. USG controls were started on day 30 (Mindray DP-10vet; 5.0 Mhz; linear probe) and USG controls were continued until day 150 in accordance with the days of blood collection. The days of fetal death/fetal loss were noted, and USG controls were continued until day 150 post-mating. Ewes that had previous pregnancies but were not pregnant at the next examination were recorded as having experienced fetal death. The study was conducted between January and June, during which the recorded minimum and maximum ambient temperatures were 3.5 °C and 24.6 °C, respectively.

Blood samples collected in January–February (A2), March–April (A4), and May–June (A6) were classified to make comparisons between the relevant periods. The sheep were divided into three groups: P+ = animals with ongoing pregnancy (*n* = 19); FD = animals with fetal death (determined after two USG controls; ewes that were pregnant in any of the controls and were determined to be non-pregnant in the subsequent control; *n* = 12); P− = animals that were determined to be non–pregnant by USG control on the 30th and 40th days of gestation and whose pregnancy did not continue and were not remated (*n* = 8).

### 2.2. Quantification of P_4_, AMH, and Hp in Sera

Quantitative determinations of serum progesterone (P_4_), anti-Müllerian hormone (AMH), and haptoglobin (Hp) were carried out using commercial ELISA kits following the detailed procedures below. All analyses were performed in accordance with Good Laboratory Practices (GLPs) to ensure data reliability and reproducibility.

Demeditec Progesterone ELISA Kit (DE1651, Lot.23K92) was used to quantitatively assess P_4_ levels in biological samples [31]. The ELISA procedure was based on the principle of competitive binding. A volume of 5 µL standard, control, and serum was pipetted into wells coated with anti-progesterone antibody. The plates were incubated for 5 min, followed by dispensing 200 µL enzyme conjugate and then incubating for 1 h. After washing 3 times, the substrate solution (TMB-200 µL) was added and incubated in the dark for 15 min. The reaction was stopped with 100 µL of stop solution, and absorbance was read at 450 nm. A standard curve was generated to calculate sample concentrations. All samples were measured in duplicate. The intra- and inter-assay coefficient of variations (%CV) were calculated as 3.06% and 6.45%, respectively. To assess linearity and accuracy, serial dilutions of high-concentration serum samples were prepared and compared with expected concentrations. Mean recovery for P4 was within the acceptable range of 92–106%, confirming assay accuracy. The results were expressed in ng/mL.

ANSHLABS AMH (Sheep) ELISA Kit (AL155, Lot.070722) was used for quantitative detection of AMH [32]. The ELISA procedure was based on a quantitative three-step sandwich-type immunoassay. A volume of 50 µL of serum and calibrators was added to each AMH-coated well, followed by 50 µL of assay buffer. The plate was incubated on an orbital microplate shaker (700 rpm) at room temperature for 2 h, washed 5 times, and then 100 µL antibody–biotin conjugate was treated. Then, the 100 µL of streptavidin-conjugate was added to the wells and incubated for 30 min, followed by 5 washes. An amount of 100 µL of TMB substrate was added and incubated at room temperature for 12 min. The reaction was stopped, and optical density was read at 450 nm. Intra- and inter-assay coefficient of variation (%CV) were calculated as 4.65% and 6.90%, respectively. Linearity was confirmed by testing serial dilutions of known AMH standards. The average recovery was 95–101%, indicating an excellent linear correlation and analytical accuracy. Results were expressed in ng/mL.

An ELISA kit (ABCAM, AB291064, Lot.10272621, Cambridge, CB2 0AX, UK) was used to quantitatively measure Hp concentration [33]. The ELISA procedure was performed following the manufacturer’s instructions, and a standard curve was generated to calculate Hp concentrations in unknown samples. Standard and serum samples (100 µL) were added to the wells and incubated for 30 min. Following incubation, a washing step was performed three times and then 100 µL enzyme–antibody conjugate was added to each well for incubating for 15 min at room temperature, followed by 3 washing steps. After multiple wash steps, TMB substrate was added, incubated for 10 min, and the reaction was terminated with 100 µL stop solution. Optical densities were recorded at 450 nm. The intra-assay and inter-assay coefficients of variation (%CV) were calculated as 4.19% and 7.00%, respectively. Linearity and accuracy checks were performed using Hp-spiked sera at different concentrations. Mean recovery was found to be between 93% and 105%, validating both analytical performance and reliability. The results were given in µg/mL.

### 2.3. Statistical Analysis

Statistical analyses were performed using IBM SPSS Statistics 27.0 (IBM Corp., New York, NY, USA). Descriptive statistics were used to calculate standard errors (±SE). The Shapiro–Wilk test was applied to assess the normality of data distribution. For non-normally distributed data, the Kruskal–Wallis test was used to detect overall group differences, followed by the Mann–Whitney U test for pairwise comparisons. The Wilcoxon test was employed for related samples that did not meet normality assumptions. For normally distributed independent samples, comparisons were made using the *t*-test. Pearson’s bivariate correlation was used to assess the strength and direction of relationships between variables. To analyze time- or period-dependent changes within subjects, the General Linear Model Repeated Measures procedure was applied. A *p*-value of <0.05 was considered statistically significant.

## 3. Results

There was a high statistically significant difference between P+ and P− animals in terms of P_4_ concentrations. P_4_ concentrations were significantly greater in P+ compared to P−, except for day 10 (*p* < 0.001; *p* < 0.0001). In terms of AMH concentrations, there was a significant difference between P+ and P− animals only on the 30th day (*p* < 0.01). Regarding the Hp concentrations, there was no difference between P+ and P− animals on the days examined (Table 1).

Although P_4_ concentrations were still high in P+ animals, they showed a significant decrease on days 30 and 40 (3.88 and 3.57 ng/mL) compared to day 10 (4.97 ng/mL), but increased significantly (5.05 and 5.77 ng/mL) from day 60 to day 80 (*p* < 0.001) and remained at the same level until day 150 (8.15 ng/mL). In P− animals, there was a significant decrease starting from day 20 of gestation compared to day 10 (*p* < 0.01), and this decrease remained at the same level until day 150 (<1 ng/mL; only at day 60: 1.70 ng/mL). In FD cases, P_4_ concentrations started to decrease starting from day 20 (1.73 ng/mL), and this decrease reached lower concentrations (*p* < 0.01; *p* < 0.001) between days 40 and 150 (1.29 ng/mL and 0.23 ng/mL; Figure 1).

In P+, AMH concentrations did not show any difference between days 10 and 60, but on the 80th and 150th gestation days, AMH concentrations decreased significantly (*p* < 0.001) compared to other days. In P− animals, AMH concentrations decreased continuously until the 80th and 150th days (*p* < 0.01), except for the increase on the 60th day. There was no difference in AMH concentrations between the days in FD (Figure 2).

After a significant increase in Hp concentrations, especially on day 20 of pregnancy (*p* < 0.001), significant decreases were observed starting from day 30 of pregnancy until the 150th day (*p* < 0.001). Similarly, Hp concentrations in FD showed the same change as in pregnancy. In P− animals, significant decreases in Hp concentrations were observed on days 80 and 150 (*p* < 0.05; Figure 3).

In terms of P_4_ concentrations, a rapid decrease (0.73 ng/mL) was observed in P− animals starting from the 20th day, while P_4_ concentrations remained above 1 ng/mL (1.73 ng/mL) in FD animals on the 20th day. Starting from the 20th day until the 40th day, the P_4_ value remained below 1 ng/mL in P− animals, while the value was above 1 ng/mL in FD animals (*p* < 0.01). On the following days, i.e., days 80 and 150, no statistically significant difference was identified between the two groups. There was no statistically significant difference between P− and FD in terms of AMH and Hp concentrations in any period (Table 2).

For the P+ group, when P_4_ concentrations were compared with FDs, there was a statistically significant difference between the two groups from day 20, and the mean concentrations of P+ were significantly greater (*p* < 0.01; *p* < 0.001). In terms of AMH, statistically significant differences were identified between P+ and FD on days 10, 20, and 30 (*p* < 0.05; *p* < 0.001). In terms of Hp, there was no significant difference between P+ and FD (Table 3).

Considering the study findings, there were statistically significant differences between A2 and A4 periods and the A2 and A6 periods regarding P_4_ in P+ animals, and the mean concentrations increased significantly during the following periods (A2: 4.46 ng/mL; A6: 8.60 ng/mL; *p* < 0.0001). In the following periods, statistically significant increases were observed in AMH serum concentrations in P+ animals in the A6 period, compared to other periods (*p* < 0.05; *p* < 0.0001). On the other hand, statistically significant decreases were observed in Hp serum concentrations in P+ animals in A4 and A6 periods (*p* < 0.01; *p* < 0.001).

In P−, there was a significant decrease in Hp concentrations in the A6 period compared to the A2 period (*p* < 0.01). However, in P− ewes, neither P_4_ nor AMH concentrations showed any change in these periods. In terms of FD, a statistically significant continuous decrease was observed in P_4_ and HP concentrations in the following periods (Table 4).

During period A2—when the body weights of the ewes were at their lowest—serum AMH concentrations, as well as Hp levels, were found to be statistically significantly greater compared to the other periods (*p* < 0.01; *p* < 0.0001; *p* < 0.05). Based on the weights of animals on day 10, AMH serum concentrations were significantly greater than the weights obtained on days 80 and 150 (*p* < 0.01; *p* < 0.001). The same result was also obtained in Hp serum concentrations, and the average value of serum concentrations taken according to the weights on day 10 was significantly greater than the weights on days 80 and 150 (*p* < 0.05; *p* < 0.01). The negative correlations observed for AMH (r = – 0.719) and Hp (r = –0.920) indicate a significant inverse relationship between body weight and these parameters (*p* < 0.01). No significant correlation was observed between P_4_ concentrations and body weight (Table 5).

When single births and twins were compared in terms of these parameters, a difference was obtained only in P_4_ concentrations. P_4_ serum concentrations of twin births were significantly greater than those of single births on days 30, 40, and 80 (*p* < 0.05; *p* < 0.01) (Figure 4).

## 4. Discussion

Historically, various studies on P_4_ changes during pregnancy and pregnancy detection in ewes have been conducted. However, the difference in P_4_ concentrations between P+ and P− ewes, the determination of embryonic deaths/losses by P_4_ concentrations, and the difference in P_4_ concentrations between single and multiple litters are still controversial [34,35,36].

This study concluded that P_4_ concentrations increased significantly between days 10 and 150. This increase was found to be significantly greater (*p* < 0.001) starting from day 60 of gestation. A study by Mukasa-Mugerwa and Viviani found that P_4_ concentrations remained at high concentrations until approximately the 20th week. The authors revealed that there was a statistical difference between P_4_ measurements made at weekly intervals (*p* < 0.001) [37]. In the same study, serum P_4_ concentrations, which averaged 8.4 ng/mL in the first trimester (day 35), increased to 13.8 ng/mL in the second trimester (day 75) and remained approximately at this value in the third trimester, with the main decrease in P4 concentrations occurring on the 3rd day of prepartum [37]. Similarly, P_4_ measurements on the days determined in our study revealed significant differences and an increase between the following days (*p* < 0.001). Therefore, P_4_ concentrations reached the highest level on days 80 and 150. P_4_ is released from the corpus luteum until approximately days 55 to 60, and after that, P_4_ is additionally released from the placenta in sheep [38,39]. The reason for the increase in P_4_ concentrations on the days in question would be the additional P_4_ produced from the placenta. The high level and capacity of luteal tissue in ewes until approximately day 142 of gestation indicates that there is a continuous steroidogenic release [40]. Kaulfuss et al. [41] also argued that this difference occurred due to the increase in P_4_ in pregnant ewes due to multiple ovulations.

The data obtained from P− animals showed P_4_ concentrations, which were 3.99 ng/mL on day 10 after mating, decreased to less than 1 ng/mL starting from day 20, and decreased to 0.26 ng/mL on day 150. Karen et al. [42] found that the average P_4_ serum value in P− animals was 0.4 ng/mL starting from day 18, and P_4_ concentrations in these animals remained between 0.2 ng/mL and 0.4 ng/mL until day 50. This difference between P+ and P− animals was also experienced in a study conducted by Al-Mousawe and Ibrahim [43]. In our study, the P_4_ value was 0.73 ng/mL on day 20 and 0.07 ng/mL and 0.32 ng/mL on days 40 and 80, supporting the findings of these authors. Statistically significant differences (*p* < 0.0001) were obtained between P+ and P− ewes on all days except day 10, including the 150th day, and it was determined that the P_4_ concentrations of P+ animals were significantly greater. Weigel et al. [44] obtained the first serum sample on the 18th day, where the P_4_ concentrations of ewes were significantly lower, and this low P_4_ level (<1.48 ng/mL) persisted until the 70th day. In this study, the mean concentrations of P_4_ in P− ewes were >1 ng/mL on the 10th and 60th days, and the concentrations were found to be <1 ng/mL on the 20th day. In a study in which both Pregnancy Associated Glycoprotein (PAG) and P_4_ serum concentrations were measured in goats, both PAG and P_4_ concentrations were significantly lower in P− ewes between 22nd and 60th days. During this period, P_4_ concentrations were found to be significantly lower in P− ewes (0.68 ng/mL on day 60 and 1.16 ng/mL on day 22) compared to P+ ewes (8.0 ng/mL and 2.8 ng/mL) [45].

Early embryonic deaths in ewes occur approximately 25–50% of the time and are caused by luteal insufficiencies, inadequate release of P_4_, and failure of the uterus to recognize pregnancy due to various factors [46]. Late embryonic deaths occur on days 25–45, while fetal losses occur after day 45 [47]. In our study, out of the 39 animals evaluated, 19 (49%) became P+, 12 (30%) experienced FD, and 8 (21%) remained P−. Among the 12 animals with detected fetal death, FD occurred in 8 animals (66.7%) on day 60, in 2 animals (16.7%) on day 80, and in another 2 animals (16.7%) on day 150. These findings indicate that the majority of fetal losses occurred during the early stages of gestation, particularly around day 60. This highlights the importance of monitoring the timing of fetal loss and enabling early diagnosis to improve reproductive performance. Our results showed that there was no statistically significant difference between P+ and FD in terms of P_4_ on day 10 after mating. On the other days when P_4_ serum concentrations were examined (starting from day 20), there was a statistically significant difference (*p* < 0.0001) between P+ and FD. However, serum P_4_ concentrations were between 1.28 ng/mL and 2.05 ng/mL in FD between days 20 and 60. Alternatively, the concentrations were determined to be <1 ng/mL on days 80 and 150. The fact that P_4_ concentrations were significantly lower than the P+ group from the beginning and decreased to basal concentrations on the days when fetal losses were determined may be an important indicator in terms of fetal losses. It was reflected that a value of ≥2.5 ng/mL was 91.4% successful in differentiating pregnancy diagnosis in P+, and this value was 98.3% and 85.3% successful in positive and negative diagnoses [34]. The fact that pregnancy concentrations were above this threshold, while FD concentrations were well below 2.5 ng/mL (<1 ng/mL), implies that these specific measurements could play a crucial role in predicting fetal losses. P_4_ measurements have been shown to be an important parameter in early or late embryonic deaths in ewes [48,49]. P_4_ plays a critical role in supporting early embryonic development, and its supplementation has been demonstrated to reduce embryonic mortality [28]. Numerous studies have reported a strong positive association between the post-ovulatory increase in P_4_ concentrations and successful embryonic development in both sheep and cattle [29,30]. In contrast, reductions in P_4_ levels have been linked to higher incidences of embryonic and fetal loss in heifers and cows [50,51]. Collectively, these findings emphasize the critical association between adequate progesterone levels and the maintenance of early pregnancy.

Statistically significant differences in P_4_ concentrations between single and twin lambs were determined at days 30, 40, and 80. Significant differences in P_4_ concentrations between single and multiple lambs have been reported in different publications [52,53,54]. They attributed the greater P_4_ value in twin or triplet pregnancies, compared to single-fetus pregnancies, to the increase in CL number. A high positive correlation was found between the number of fetuses, CL diameter, and P_4_ concentrations [55]. In our study, there was no difference between singletons and twins after day 80, which may be related to the release of other placental hormones, besides P_4_, during this period. Research has demonstrated that placental P_4_ production initially increases between 50 and 70 days of gestation, followed by a second surge between 90 and 120 days. These hormonal changes after day 80 may have made it difficult to distinguish between singletons and twins.

In consideration of the study results, P_4_ was statistically different between A2 and A4 months, and between A2 and A6 periods in P+ animals, with concentrations increasing significantly in the following months. According to a study conducted between June–September and March–May, P_4_ concentrations were found to be significantly greater in June–September than in March–May [56]. Since the highest P_4_ concentrations were obtained in May–June, our study reinforces the idea that these concentrations may be related to suitable air temperatures and the comfortable shelter conditions offered, which vary with feeding practices and periods from region to region. For example, Spencer and Bazer [57] revealed that prolactin hormone peaks in mother ewes starting from day 50 until day 120–130 of pregnancy. The results indicate that serum P_4_ concentrations increase as pregnancy progresses, and this rise may also be influenced by seasonal factors. In non-pregnant ewes, no significant changes in P_4_ concentrations were observed during the same periods, likely due to their anestrous status. Moreover, the occurrence of parturition in both the A4 and A6 groups suggests that this variation in P_4_ concentrations is not directly related to the advancing stage of pregnancy.

AMH and Hp serum concentrations decreased significantly in both P+ and P− sheep, especially on days 80 and 150 compared to the initial data (*p* < 0.05; *p* < 0.0001). There were no changes in AMH serum concentrations in FDs in the following days. AMH forms a wave-like dynamic profile during pregnancy. Immunohistochemistry studies have proven that AMH is expressed in healthy preantral and antral follicles on day 60 of gestation. Intrafollicular AMH concentration was not different in pregnant and non-pregnant cows on day 60. In this study, generally, no difference was found in AMH serum concentrations on all days except day 30. Both the variation of AMH serum concentrations between days and the difference between P+ and P− on day 30 may be attributed to the low number of follicles releasing high concentrations of AMH [14]. In non-fertile ewes, the number of follicles is lower, resulting in lower AMH blood concentrations [58]. It appears that variations in follicle numbers during different stages of gestation may lead to significant differences in serum hormone levels between pregnant and non-pregnant ewes. The increase in Hp values in obesity suggests that there is a significant link between Hp and weight [59]. Serum Hp has also been reported to be an important indicator of weight gain in pigs [60].

Interestingly, there were significant statistical differences (*p* < 0.05; *p* < 0.001) in AMH concentrations between P+ and FD on days 10, 20, and 30. There are very few publications on AMH in relation to pregnancy status, which leads to controversial results [61]. The results have shown that AMH is positively correlated with the number of histologically determined primordial, transitory primary, secondary, and antral healthy follicles [62]. Our findings show that AMH serum concentrations were significantly greater in P+ ewes from the 10th day until the 30th day compared to the FD group. The fact that high intrafollicular P_4_ and AMH concentrations affect fertility also supports our findings in this study [61]. Increased AMH concentrations in goats result in an increase in total CL number, embryo number, and embryo quality, demonstrating the hormone’s role in embryo development [63].

An interesting finding in terms of AMH is that the concentrations in P+ ewes increased to the highest level in A4. The increase of the AMH serum value from 1.91 ng/mL in A4 months to 8.15 ng/mL in A6 months (*p* < 0.0001) is a result that should be taken into consideration and reveals that AMH concentrations change according to different months/seasons [64]. This result can be explained by the provision of quality grass and feed in May-June and the ability of animals to move outside to pasture.

The absence of any seasonal change in the P− and FD groups may be related to the decrease in the number and quality of follicles and follicle degeneration during this period (A6) [65]. Evaluating pregnancy rates due to seasonal variations is out of the scope of the present study.

Within the framework of this study, AMH concentrations decreased significantly with increasing sheep body weight compared to low body weight. In a study conducted in dogs, there were statistically significant differences in AMH serum concentrations between small, medium, and large giant breeds [66]. Likewise, the findings of Evci et al. [67] showed AMH concentrations were high in dogs weighing < 10 kg, while the concentrations decreased significantly in dogs weighing 25 kg. Such a trend of decreasing AMH concentrations may be a result of FSH release and changes in follicle numbers due to weight gain; this decrease in AMH concentrations due to FSH release was also reported by Monniaux et al. [14]. Additionally, the relationship between this decrease in AMH concentrations and subclinical metabolic diseases should be examined in further study. In terms of Hp, the same course was observed in P+ and P− sheep in the following days. On days 80 and 150, significant decreases were observed in both groups compared to the initial concentrations. In a study conducted in mares [68], Hp concentrations gradually increased from the 5th to the 10th month of pregnancy (*p* < 0.05), and started to decrease in the 11th month, which is consistent with our Hp value decreases towards the end of pregnancy. The fact that Hp concentrations reached low concentrations in both FD and P− animals across all the days examined in the study indicates that Hp measurements are not an important parameter for FD discrimination. This situation was the same for P− and P+, and there was no significant difference between P− and P+. Hp concentrations showed a decreasing trend in all groups (P+, P−, and FD). Therefore, seasons did not have a different effect on Hp in these groups. The lack of variation in P_4_ concentrations throughout the A2–A6 period suggests that the observed changes in AMH and Hp concentrations are more likely associated with body weight. Additionally, the strong negative correlations found between body weight and AMH (r = –0.719), as well as Hp (r = –0.920), further emphasize this relationship (*p* < 0.01). In contrast, no significant correlation was observed between body weight and P_4_ concentrations.

Hp has been shown to act as an indicator in various infectious diseases [69]. Significant increases in Hp concentrations were observed in cases of pregnancy toxemia compared to normal healthy ewes [70,71,72]. These data show that Hp concentrations are more influenced during infections or metabolic disorders.

As in AMH value, this study noted that significant decreases occurred in Hp serum concentrations as ewe weights increased (*p* < 0.05; *p* < 0.01). On the other hand, Hp concentrations showed significant decreases in the period two weeks before delivery [73]. In pigs, a high correlation between body weight gain and Hp serum concentrations was also identified [60]. Free hemoglobin has toxic and oxidative effects in the blood. By binding to hemoglobin, the formation of oxygen radicals is prevented by the effect of iron, and thus, oxidative damages through hemolysis are eliminated. Thus, Hp plays a role in the antioxidant system [73]. Hp also has a bacteriostatic effect because it binds iron and inhibits bacterial growth by binding iron [74]. Apparently, in the following days, body weight gain (depending on the growth of the offspring or pups) leads to a decrease in Hp concentrations as the sheep’s mechanism protects itself due to the above-mentioned mechanism.

## 5. Conclusions

In conclusion, this study found that the pregnancy process can be monitored based on the increases in P_4_ obtained depending on the days, while the P+ and P− ewes can be determined from day 20 of pregnancy since FD can be revealed with low P_4_ serum results obtained compared to P+ ewes. Moreover, P_4_ concentrations are significantly lower in ewes with low weight, concluding that special attention should be given to these animals. Determination of single and twin pregnancies from P_4_ concentrations should be considered in terms of nutrition and protection against metabolic diseases. With regard to the AMH, the significant differences between P+ and FD from day 10 to day 30 suggest that this may be related to the poorer follicle quality of FD compared to the P+ group. Although the mean AMH concentrations were low (statistically not significant) in P− animals from day 20 onwards, a difference between P+ and P− animals was observed only on day 30, suggesting that there may be a relationship between AMH concentrations and the continuation of pregnancy. The fact that P_4_ and AMH concentrations were high in the A6 period suggests that greater productivity can be obtained by synchronization in ewes and intensifying ram siring in these months. The significant decrease in AMH concentrations in ewes with high body weight in the later gestation days (150th day), together with other hormonal data, indicates that it may be an important parameter for impending parturition. The decreases in Hp concentrations due to the increase in body weight indicate that the body regulates its own defense mechanism.

## Figures and Tables

**Figure 1 vetsci-12-00463-f001:**
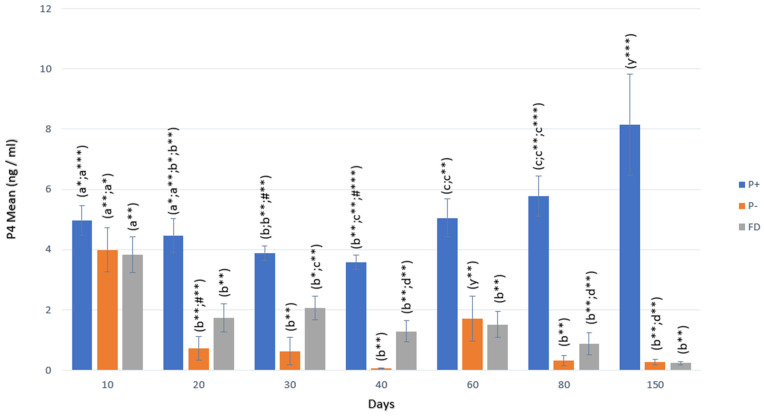
P_4_ change in P+, P−, and FD by days. Different letters or asterisk indicate differences between days (a:b:c:*; *p* < 0.05; a:b:c:d:#**; *p* < 0.01; a:b:c:y:#***; *p* < 0.001).

**Figure 2 vetsci-12-00463-f002:**
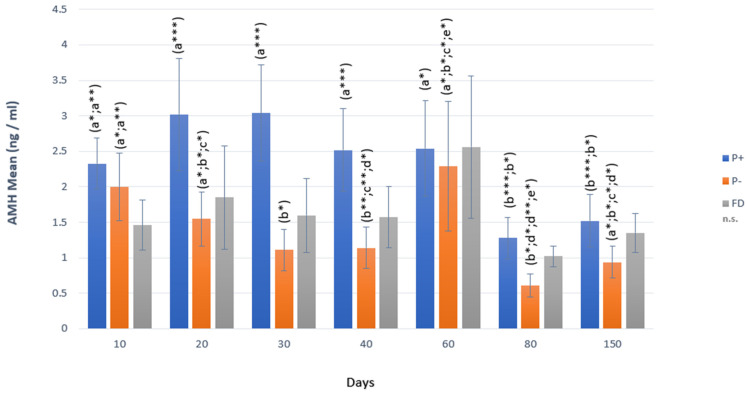
AMH changes in P+, P−, and FD by days. Different letters or asterisks indicate differences between days (a:b:c:d:e*; *p* < 0.05; a:b:c:d**; *p* < 0.01; a:b***; *p* < 0.001).

**Figure 3 vetsci-12-00463-f003:**
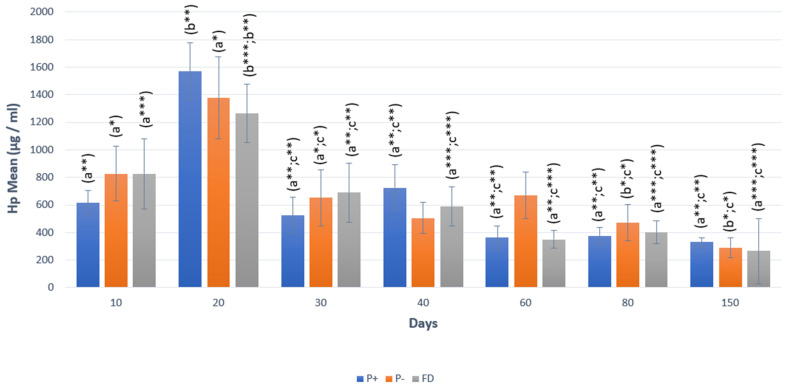
Hp changes in P+, P−, and FD by days. Different letters or asterisk indicate differences between days (a:b:c*; *p* < 0.05; a:b:c**; *p* < 0.01; a:b:c***; *p* < 0.001).

**Figure 4 vetsci-12-00463-f004:**
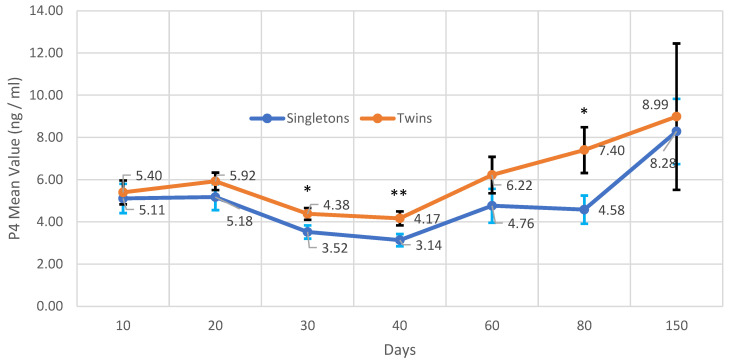
Comparison of P_4_ concentrations in single and twin pregnancies (single offspring; *n* = 11); twin offspring (*n* = 8). Different asterisks indicate differences between days (* *p* < 0.05; ** *p* < 0.01).

**Table 1 vetsci-12-00463-t001:** Difference in P_4_, AMH, and Hp Among P+ and P− ewes. Results are presented in Mean ± SE.

	P_4_ (ng/mL)			AMH (ng/mL)			Hp (µg/mL)		
Days	P+	P−	*p*	P+	P−	*p*	P+	P−	*p*
10	4.97 ± 0.50	3.99 ± 0.74	n.s.	2.33 ± 0.36	2.00 ± 0.48	n.s.	613.92 ± 88.46	827.42 ± 196.72	n.s.
20	4.46 ± 0.57 ^(a^***^)^	0.73 ± 0.39 ^(b^***^)^	*** *p* < 0.0001	3.02 ± 0.79	1.55 ± 0.38	n.s.	1570.51 ± 206.54	1378.27 ± 296.17	n.s.
30	3.88 ± 0.23 ^(a^***^)^	0.63 ± 0.46 ^(b^***^)^	*** *p* < 0.0001	3.04 ± 0.68 ^(s^**^)^	1.11 ± 0.29 ^(b^**^)^	*p* < 0.01	523.13 ± 131.71	650.83 ± 202.15	n.s.
40	3.57 ± 0.24 ^(a^***^)^	0.07 ± 0.01 ^(b^***^)^	*** *p* < 0.0001	2.52 ± 0.58	1.14 ± 0.29	n.s.	724.33 ± 166.61	505.48 ± 110.85	n.s.
60	5.05 ± 0.63 ^(a^**^)^	1.70 ± 0.75 ^(b^**^)^	** *p* < 0.001	2.54 ± 0.67	2.29 ± 0.91	n.s.	362.65 ± 85.75	671.17 ± 167.98	n.s.
80	5.77 ± 0.67 ^(a^***^)^	0.32 ± 0.16 ^(b^***^)^	*** *p* < 0.0001	1.28 ± 0.29	0.61 ± 0.16	n.s.	375.72 ± 60.59	471.86 ± 130.36	n.s.
150	8.15 ± 1.67 ^(a^***^)^	0.26 ± 0.09 ^(b^***^)^	*** *p* < 0.0001	1.52 ± 0.37	0.94 ± 0.22	n.s.	329.65 ± 31.65	289.90 ± 72.63	n.s.

Different letters or asterisks in rows indicate differences between days (** *p* < 0.01; *** *p* < 0.0001). n.s. means “statistically no significant difference”.

**Table 2 vetsci-12-00463-t002:** Comparison P_4_, AMH, and Hp concentrations in P− ewes and FD on different days. Results are presented in Mean ± SE.

	P_4_ (ng/mL)			AMH (ng/mL)			Hp (µg/ml)		
Days	P−	FD	*p*	P−	FD	*p*	P−	FD	*p*
10	3.99 ± 0.74	3.83 ± 0.59	n.s	2.00 ± 0.48	1.46 ± 0.35	n.s	827.42 ± 196.72	826.76 ± 255.12	n.s
20	0.73 ± 0.39 ^(a^*^)^	1.73 ± 0.4 ^(b^*^)^	*p* < 0.05	1.55 ± 0.38	1.85 ± 0.73	n.s	1378.27 ± 296.17	1262.95 ± 211.74	n.s
30	0.63 ± 0.46 ^(a^**^)^	2.06 ± 0.40 ^(b^**^)^	*p* < 0.01	1.11 ± 0.29	1.60 ± 0.52	n.s	650.83 ± 202.15	688.91 ± 214.04	n.s
40	0.07 ± 0.01 ^(a^**^)^	1.29 ± 0.36 ^(b^**^)^	*p* < 0.01	1.14 ± 0.2	1.57 ± 0.43	n.s	505.48 ± 110.85	586.84 ± 142.42	n.s
60	1.70 ± 0.75 ^(a^**^)^	1.51 ± 0.43 ^(b^**^)^	*p* < 0.01	2.29 ± 0.91	2.56 ± 1.00	n.s	671.17 ± 167.98	350.08 ± 64.88	n.s
80	0.32 ± 0.16	0.88 ± 0.37	n.s	0.61 ± 0.16	1.02 ± 0.15	n.s	471.86 ± 130.36	399.90 ± 82.29	n.s
150	0.26 ± 0.09	0.23 ± 0.06	n.s	0.94 ± 0.22	1.35 ± 0.27	n.s	289.90 ± 72.63	264.96 ± 34.59	n.s

Different letters in rows indicate the differences between groups (* *p* < 0.05; ** *p* < 0.01). n.s. means “statistically not significant”.

**Table 3 vetsci-12-00463-t003:** Comparison of P_4_, AMH, and Hp concentrations on different days of pregnancy and fetal deaths. Results are presented in Mean ± SE.

	P_4_ (ng/mL)			AMH (ng/mL)			Hp (µg/mL)		
Days	P+	FD	*p*	P+	FD	*p*	P+	FD	*p*
10	4.97 ± 0.50	3.83 ± 0.59	n.s	2.33 ± 0.36 ^(a^*^)^	1.46 ± 0.35 ^(b)^	n.s	613.92 ± 88.46	826.76 ± 255.12	n.s
20	4.46 ± 0.57 ^(a^**^)^	1.73 ± 0.47 ^(b^**^)^	** *p* < 0.01	3.02 ± 0.79 ^(a^*^)^	1.85 ± 0.73 ^(b)^	n.s	1570.51 ± 206.54	1262.95 ± 211.74	n.s
30	3.88 ± 0.23 ^(a^***^)^	2.05 ± 0.40 ^(b^***^)^	*** *p* < 0.001	3.04 ± 0.68 ^(a^*^)^	1.60 ± 0.52 ^(b^*^)^	* *p* < 0.01	523.13 ± 131.71	688.91 ± 214.04	n.s
40	3.57 ± 0.24 ^(a^***^)^	1.28 ± 0.36 ^(b^***^)^	**** *p* < 0.0001	2.52 ± 0.58	1.57 ± 0.43	n.s	724.33 ± 166.61	586.84 ± 142.42	n.s
60	5.05 ± 0.63 ^(a^***^)^	1.51 ± 0.43 ^(b^***^)^	**** *p* < 0.0001	2.54 ± 0.67	2.56 ± 1.00	n.s	362.65 ± 85.75	350.08 ± 64.88	n.s
80	5.77 ± 0.67 ^(a^***^)^	0.88 ± 0.37 ^(b^***^)^	**** *p* < 0.0001	1.28 ± 0.29	1.02 ± 0.15	n.s	375.72 ± 60.59	399.90 ± 82.29	n.s
150	8.15 ± 1.67 ^(a^***^)^	0.23 ± 0.06 ^(b^***^)^	**** *p* < 0.0001	1.52 ± 0.37	1.35 ± 0.27	n.s	329.65 ± 31.65	264.96 ± 34.59	n.s

Different letters in rows indicate the differences between groups (* *p* < 0.05; ** *p* < 0.01; *** *p* < 0.001; **** *p* < 0.0001). n.s. means “statistically not significant”.

**Table 4 vetsci-12-00463-t004:** Changes in P_4_, AMH, and HP concentrations in different months. Results are presented in Mean ± SE.

	P_4_ (ng/mL)			AMH (ng/mL)			Hp (µg/ml)		
Months	P+	P−	FD	P+	P−	FD	P+	P−	FD
A2	4.46 ± 0.19 ^(a^***^)^	1.35 ± 0.3 ^(a)^	2.23 ± 0.26 ^(a^**^,a^***^)^	2.52 ± 0.29 ^(a^*^, a^***^)^	1.5 ± 0.19 ^(a)^	1.62 ± 0.26 ^(a)^	857.97 ± 89.78 ^(a^***^)^	840.50 ± 116.90 ^(a;b^**^)^	841.37 ± 108.23 ^(a^*^)^
A4	5.56 ± 0.45 ^(b^***^)^	1.01 ± 0.41 ^(a)^	1.20 ± 0.29 ^(b^**^)^	1.91 ± 0.38 ^(b^*^)^	1.5 ± 0.50 ^(a)^	1.79 ± 0.52 ^(a)^	369.19 ± 51.79 ^(b^***^)^	571.52 ± 105.88 ^(a)^	374.99 ± 51.51 ^(b^*^)^
A6	8.60 ± 1.71 ^(b^***^,c^**^)^	0.26 ± 0.09 ^(a)^	0.23 ± 0.06 ^(b^***^)^	8.15 ± 1.67 ^(b^*^,c^***^)^	0.9 ± 0.22 ^(a)^	1.36 ± 0.27 ^(a)^	329.65 ± 31.65 ^(b^***^, c^**^)^	289.90 ± 72.63 ^(c^**^)^	264.96 ± 34.59 ^(b^*^; c^*^)^

Different letters in columns indicate statistical difference (* *p* < 0.05; ** *p* < 0.001; *** *p* < 0.0001).

**Table 5 vetsci-12-00463-t005:** Changes in P_4_, AMH, and Hp concentrations according to body weight measured on different days in pregnant ewes. Results are presented in Mean ± SE.

Weight Measurement Days (days)	Weights (kg)	P_4_ (ng/mL)	AMH (ng/mL)	Hp (µg/mL)
10	67.26 ± 6.71	4.97 ± 0.50 ^(a^*^)^	2.39 ± 0.36 ^(a^***^;a^**^)^	613.91 ± 88.46 ^(a^*^;a^**^)^
80	76.18 ± 8.98	5.77 ± 0.66 ^(a^*^)^	1.27 ± 0.29 ^(b^***^)^	375.72 ± 60.58 ^(b^*^)^
150	86.15 ± 9.14	8.15 ± 1.67 ^(a^*^)^	1.52 ± 0.37 ^(b^**^)^	329.64 ± 31.64 ^(b^*^;b^**^)^
*p*	*p* > 0.05	*p* > 0.05	a**:b** *p* < 0.01 a***:b*** *p* < 0.001	a*:b* *p* < 0.05 a**:b** *p* < 0.01

Different letters in columns indicate statistical difference (* *p* < 0.05; ** *p* < 0.001; *** *p* < 0.0001).

## Data Availability

The data that support the findings of this study are available from the corresponding authors upon reasonable request.

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
