# Peer review of "Concentrations of Progesterone (P4), Anti-Müllerian Hormone (AMH), and Haptoglobin (Hp) in Pregnant and Non-Pregnant Ewes and Their Association with Fetal Mortality, Maternal Weight, and Twinning Rate"

_vetsci, 2025, doi:10.3390/vetsci12050463_

Round 1

Reviewer 1 Report

Comments and Suggestions for Authors

The study is well justified and of interest for the scientific community. This reviewer has the following concerns and suggestions:

  1. Introduction and Results are meaningful and have an appropriate length, Discussion should be shorter. Avoid the repitition of results in the Discussion. When comparing exact concentration/values from different studies, please keep in mind that different methods or test kits are hardly comparable, in particular when analyses were performed many years ago, e.g., Weigel et al, 1981. Thus, this reviewer recommends to discuss trends, thresholds or tendency rather than exact concentrations.
  2. Please discuss limitations of your study, e.g., limited number of animals.
  3. Line 357 and following: This reviewer does not agree that AMH "caused" low pregnancy rates or "increased" PR or "affects fertilty". AMH may indicate a lower or greater risk for pregnancy, but cannot be regarded as causative, in particular as animals were already pregnant at the time of AMH determination. Please rephrase.
    If the authors see a causative mechanism (not only a correlation) for AHM produced in follicles of pregnant animals and non-pregnant animals, please feel free to speculate. Same for the rise of AMH concentration in P- and FD animals at d 60.

General editorial comments

  1. This reviewer suggests to use "concentrations" rather than "values" or "levels".
  2. Concentrations or values are "greater" not "higher". “Higher” and “lower” refer to three-dimensional space. Substitute greater, lesser.
  3. Rethink if concentration should be presented with two decimals. With regard to the limited number of animals and the spread of data (SE), the second decimals provides limited information.  
  4. Table 5: Add units (kg, ng/mL, etc)
  5. The manuscript should be checked by a professional proofreader.

Reviewer 2 Report

Comments and Suggestions for Authors

The present work examines the changes in haptoglobin, anti-Müllerian hormone, and progesterone values in pregnant, non-pregnant, and subjects undergoing fetal loss in sheep, revealing the relationships between these parameters and phenomena such as maternal weight and twins. This approach stems from the need to find usefull markers to assess the progression of pregnancy in sheep. The topic is relevant because it provides insights into the field of the endocrinology of sheep reproduction. The references are specific, however, in some parts, the discussion could be further developed. I suggest adding some pertinent references. I encourage you to improve the quality of the English used in the manuscript, particularly in the first part, where the language lacks fluency. I assume this may be due to the fact that different sections were written by different authors.

Simple Summary: This section should contain a clear statement of the problem addressed, the aims and objectives, pertinent results, conclusions from the study. It should be no more than 200 words. I think that this section should be rewritten.

Abstract: They recap the information contained in the main text, but it does not properly include the background, while it is necessary to place the addressed question in a broader context. Also, Line 28: specify, even it is obvious, that you are referring to the pregnant animals. Line 31, Lines 36-37: fix these sentences, they contain some errors. Please review the text carefully, because the sentences are difficult to understand and the text lacks fluency. Moreover, the abstract is too long, as according to the journal's guidelines it should contain approximately 200 words. When rewriting this paragraph, please keep this in mind.

Keywords: I suggest you to add some relevant words: "Abortion", “Sheep”, “Twin gestation”, “Singleton gestation”, and “Body condition”, because they are pertinent to the application of your results.

Title and Introduction:  The title is attractive and adequate for the paper's content, but you should add “mother weight” and eliminate the P4 acronym, because it is not necessary here (it is not a protein or gene), and explain its meaning the first time it appears. The introduction  partially contextualizes the main aspects of the topic. I believe you should further elaborate on the relationship between the measured parameters and what was carried out in your study. For example, regarding haptoglobin, its supposed role in pregnancy or abortion is not discussed. As for progesterone, there is no reference to its levels in twin versus singleton pregnancies, or about abortion. Concerning AMH, its potential role or significance during pregnancy has not been addressed, so it is unclear to the reader why it should be considered a marker. You should include these details to convince the reader that your parameters may play a role as markers in predicting events that may occur during pregnancy. The aim is expressed, however, the hypothesis should be clearly formulated.

Materials and Methods: Animals, examinations and sampling are properly presented, but I think you should add more information about the animals (mean of the body weight at the beginning of the study, if they were nulliparous, primiparous or multiparous, the mean of the age with the SD, breed). Were they receiving any medication during the study, or did they have any underlying medical conditions? Where has the study been conducted? It would be valuable to include details about the month in which the study took place, along with the average environmental temperature and humidity. All this information is crucial in a physiological and endocrinological study. In the statistical analysis, you should better express which test you used to evaluate between-period differences in the different groups with the subsequent post-hoc test and the test to assess the differences between groups within each period. Why did not you take the sex of the fetus into consideration? It would be interesting.

Results: They are logically presented and accompanied by clear tables.

Discussion and Conclusion: The discussion follows a logical line, developing persuasive interpretations. However, the current knowledge on the topic could be further summarized adding some references to make your interpretations more solid. In particular, for the Hp, I suggest delving deeper into the subject, explaining the putative role of Hp during pregnancy.  It would be appropriate to include a comparison with other species, such as the horse, in which some of the parameters you investigated have already been studied. Below are some proper references:

  • Satué, K., Marcilla, M., Medica, P., Ferlazzo, A., & Fazio, E. (2018). Sequential concentrations of placental growth factor and haptoglobin, and their relation to oestrone sulphate and progesterone in pregnant Spanish Purebred mare. Theriogenology115,77–83. https://doi.org/10.1016/j.theriogenology.2018.04.033

The limit of your study should be discussed.

Reviewer 3 Report

Comments and Suggestions for Authors

Thank you for the opportunity to review this manuscript. Please consider the following points:

Introduction

  • I think the paper would be strengthened if there were a few more sentences describing the problem that is trying to be solved (e.g., identifying ways to determine pregnancy earlier than 40 days? dentifying twins with blood markers?)
  • If the correlation between the markers and birthweight or the number of offspring per pregnancy are going to be investigated, perhaps include a bit more information in the background on the topic, and why it is important to explore the relationships between birthweight, number of offspring, and P4, AMH, and Hp.

Methods

  • Lines 107-113: When were the sheep bred, and were they all bred at the same time? Were the sheep synchronized prior to breeding? This information is important to include, as it confirms that the A2, A4, and A6 groupings include sheep that are at a similar point in pregnancy.
  • Lines 107-114: the total group was 40 animals, but the sum of 19, 12, and 8 is only 39. What happened to the other one?

Results

  • Line 143: which two comparisons do the two cited p values reference? In reality, readers may get more from this sentence if the direction of the comparison is emphasized with the p values. For example, you could word it something like this: “Serum P4 was significantly increased in P+ animals compared to P- animals except for D10 (p < 0.001)”.
  • Lines 217-221: I think this needs to be reworded. I do not think that it can be stated that the AMH and Hp levels are "based on weight”, because the independent variable is the day, not the weight. Instead, it perhaps could be stated that there is a correlation between weight and the blood markers (if there was one based on the stats).

Discussion

  • Lines 327-337: How can the increase in P4 be attributed to seasons and not to the growing pregnancy? How can the two be separated, especially if there is no seasonal change of P4 in the P- group?
  • Lines 440-441: the change in Hp cannot be attributed to a change in BW, as the change in BW is not the independent variable
  • Lines 436-438: Perhaps I am missing something here. What time of year were the sheep pregnant? In other words, had they already lambed before the start of the A2 period, or were they pregnant throughout the A2, A4, and A6 timeframes? Please include information about when the sheep were bred, and when they lambed. This way, readers can understand the meaning of the data from the A2-A6 periods. If the sheep were not pregnant during this time, and there was a difference in P4 levels in the group of sheep that were in the P+ and P- groups, then the change in P4 is an interesting observation for sure. However, if the P+ group was still pregnant during the A2-A6 timeframes, then changes in P4 cannot be attributed to season alone, as these animals were pregnant and increases in P4 could be due to the growing pregnancy. Based on the P4 levels presented, it appears that the sheep were indeed pregnant during the A2-A6 periods.

Table 1

  • For P4 on D60, the table states that **p < 0.0001, however the table title describes ** as representing p < 0.001. Please double check and correct where needed.

Figures 1-3

  • Consider splitting each of these up into three parts or three different graphs. Currently, the overlap of the treatment groups makes it a bit confusing to look at the data. For example, on D60 of AMH, it is difficult to discern which group (P+ or FD) has the a*.
  • Also, it is difficult to discern which comparisons belong to the **. For example, on D10 of the P4 graph, the P+ group has both an “a*” and an “a***”. Which comparison is described 

Reviewer 4 Report

Comments and Suggestions for Authors This paper analyzes changes in progesterone, haptoglobin, and anti-Müllerian hormone concentrations in pregnant, non-pregnant, and embryonic death ewes. It represents a significant contribution to the study of pregnancy in sheep, but some changes need to be made for publication. In the introductory paragraph, lines 63 to 73, please clarify whether the data presented are from sheep. I believe it would be important in this introduction to base this on results from other species, as they suggest that the molecules they are seeking to detect may be related to embryonic death. The materials and methods are well described and appropriate for the work presented. In the results, it would be important to explain why the fetal death group is not included in Table 1. In the discussion, as was the case in the introduction, the relationship of haptoglobin to the reproductive process is unclear. This aspect should be explored in more detail. The conclusions are appropriate and summarize the most important contributions of the paper.

Reviewer 5 Report

Comments and Suggestions for Authors

In their current manuscript, H.G. Ozturan et al. investigate the serum concentrations of progesterone (P4), anti-Müllerian hormone (AMH), and haptoglobin (Hp) in pregnant and non-pregnant Assaf ewes using ELISA assays. The study also attempts to correlate these hormonal parameters with foetal mortality, birth weight, and twinning rate. The introduction is generally well written and emphasizes the global significance of sheep production. However, it lacks clarity regarding the specific country in which the study was conducted. Relevant and recent data on sheep production in Turkey, or the country where the research took place, should be incorporated.

In the Materials and Methods section, while the quantification of P4, AMH, and Hp appears to follow standard protocols, it is insufficient to merely state that procedures were carried out according to manufacturers’ instructions. Additional methodological details are necessary to ensure reproducibility. Moreover, the opening sentences (lines 97 and 98) require clearer articulation. The statement that the average age of the animals was three years and that all animals were of the same age appears redundant. If there is an “average age” the animals are not all of the same age (3 years)- Furthermore, the reproductive history of the ewes is not addressed and should be specified.

The study reiterates known findings, such as the significant distinction in P4 levels between pregnant and non-pregnant ewes, but also presents more novel observations. For example, P4 concentrations were significantly higher in twin pregnancies compared to singleton pregnancies on days 30, 40, and 80. Additionally, AMH levels were elevated in pregnant ewes compared to those that experienced early foetal loss, while differences in Hp levels between pregnant and non-pregnant animals were not statistically significant. The seasonal variation in P4 and AMH concentrations is also noteworthy. However, the geographical location of the study must be clearly stated to contextualize these findings.

The discussion is generally well-structured, though certain points warrant more thorough examination. For instance, the sample size and timing of foetal loss within the foetal mortality group should be described in greater detail. Moreover, the potential impact of hormone levels on birth outcomes, such as survival to term and neonatal weight, should be explored more comprehensively.

Round 2

Reviewer 4 Report

Comments and Suggestions for Authors

In the present form the manuscript can be accepted